# Risk of psychiatric disorders among the surviving twins after a co-twin loss

**Huan Song[1,2,3]\*, Henrik Larsson[3,4], Fang Fang[5], Catarina Almqvist[3,6], Nancy L Pedersen[3,7], Patrik KE Magnusson[3], Unnur A Valdimarsdóttir[2,3,8]**

[1]West China Biomedical Big Data Center, West China Hospital, Sichuan University, Chengdu, China; [2]Center of Public Health Sciences, Faculty of Medicine, University of Iceland, Reykjavík, Iceland; [3]Department of Medical Epidemiology and Biostatistics, Karolinska Institutet, Stockholm, Sweden; [4]School of Medical Sciences, Örebro University, Örebro, Sweden; [5]Institute of Environmental Medicine, Karolinska Institutet, Stockholm, Sweden; [6]Astrid Lindgren Children's Hospital, Karolinska University Hospital, Stockholm, Sweden; [7]Department of Psychology, University of Southern California, Los Angeles, United States; [8]Department of Epidemiology, Harvard TH Chan School of Public Health, Boston, United States

**Abstract** Losing a co-twin by death is a severely stressful event yet with unknown impact on the surviving twin's risk of psychiatric disorders. We identified all Swedish-born twins who lost a co-twin by death between 1973 and 2013 (n = 4,528), their 4939 non-twin full siblings, together with 22,640 age- and sex-matched non-bereaved twins. Compared to the non-bereaved twins, exposed twins were at increased risk of receiving a first diagnosis of psychiatric disorders (hazard ratio = 1.65, 95% confidence interval1.48–1.83), particularly during the first month after loss. Similarly, compared to non-twin full siblings, the relative risks were significantly increased after loss of monozygotic co-twin (2.45-fold), and loss of a dizygotic co-twin (1.29-fold), with higher HR observed with greater age gaps between twins and non-twin siblings. As dizygotic twins share equal genetic relatedness to the deceased twin as their full siblings, this pattern suggests that beyond the contribution of genetic factors, shared early life experiences and attachment contribute to the risk of psychiatric disorders among surviving twins after co-twin loss.

\*For correspondence:
songhuan@wchscu.cn

**Competing interests:** The authors declare that no competing interests exist.

## Introduction

The death of a close relative has been linked to a variety of adverse health consequences among bereaved family members. An excess risk of morbidity and mortality was observed among individuals who experienced the loss of a spouse (*Prior et al., 2018*; *Brenn and Ytterstad, 2016*), parent (*Bylund Grenklo et al., 2013*; *Rostila et al., 2016*), or child (*Hendrickson, 2009*; *Li et al., 2003*). Nevertheless, although the sibling tie represents one of the most enduring kin relations across the life course (*Bedford, 2012*), sibling loss has received relatively scant attention in the scientific literature. A recent Danish population-based study compared the relative weight of the different types of bereavement (loss of child, spouse, sibling, and parents) with regard to the subsequent risk of suicide, deliberate self-harm and psychiatric illness (*Guldin et al., 2017*). The results revealed that, although loss of a sibling led to less obvious short-term rate elevations (compared to the loss of a child or spouse), sibling bereavement was associated with the greatest long-term risk elevations of the studied adverse psychiatric consequences.

By virtue of being born at the same time, both monozygotic and dizygotic twins tend to have more in common (e.g., early life experiences) and report stronger emotional bond than other sibling-pairs (*Bank and Kahn, 1997*; *Neyer, 2002*). Moreover, the 100% genetic share between monozygotic twins may contribute to even closer emotional ties between them than dizygotic

twins (*Neyer, 2002*; *Cassell, 2011*; *Fortuna et al., 2010*). Consequently, losing a co-twin by death may be a particularly devastating life stressor with considerable health implications for the surviving twins (*Segal and Bouchard, 1993*). Nevertheless, scientific studies on this topic are relatively rare. With regard to psychiatric reactions to a co-twin death, only a handful of descriptive studies with small sample sizes have been conducted, mainly using self-reported measures of grief (i.e. sorrow or general sadness) as an outcome (*Segal et al., 1995*; *Segal and Ream, 1998*; *Rosendahl and Björklund, 2013*). Evidence on a potential rise in rates of clinically diagnosed psychiatric disorders associated with the loss of a co-twin is, however, totally lacking. Moreover, given the shared genetic background of twins, the psychiatric consequences after a co-twin loss need to be estimated with caution (*Segal, 2009*; *Pompili et al., 2006*), ideally using family comparison with additional vigorous control of previous psychiatric morbidities and the causes of the co-twin's death.

Due to the sharp rise of medically assisted reproduction and delayed childbearing, the twinning rate, particularly the rate for dizygotic twins, has increased dramatically in all developed countries since 1970 s (*Pison et al., 2015*) which calls for a vigorous documentation of the potential health consequences suffered by surviving twins after a co-twin loss. Therefore, taking advantage of nationwide health registers in Sweden and the unique Swedish Twin Registry, we conducted a nationwide population- and sibling-matched cohort study to estimate the short- and long-term association between loss of a co-twin and subsequent risk of first-recorded psychiatric disorders.

## Results

Here, we included all Swedish-born twins that lost a co-twin by death between 1973 and 2013 (exposed twins), their full siblings, together with a group of age- and sex-matched unexposed twins who were randomly selected from the twin population and did not experience such a loss. While comparing twins exposed to a co-twin loss to twins without such an experience allowed control for pregnancy-, birth- and some psychosocial aspects of being a twin (the matched twin cohort), the comparison of exposed twins with their full siblings within the same family enabled a direct comparison between two types of stressors, that is loss of a co-twin and loss of a full sibling, while

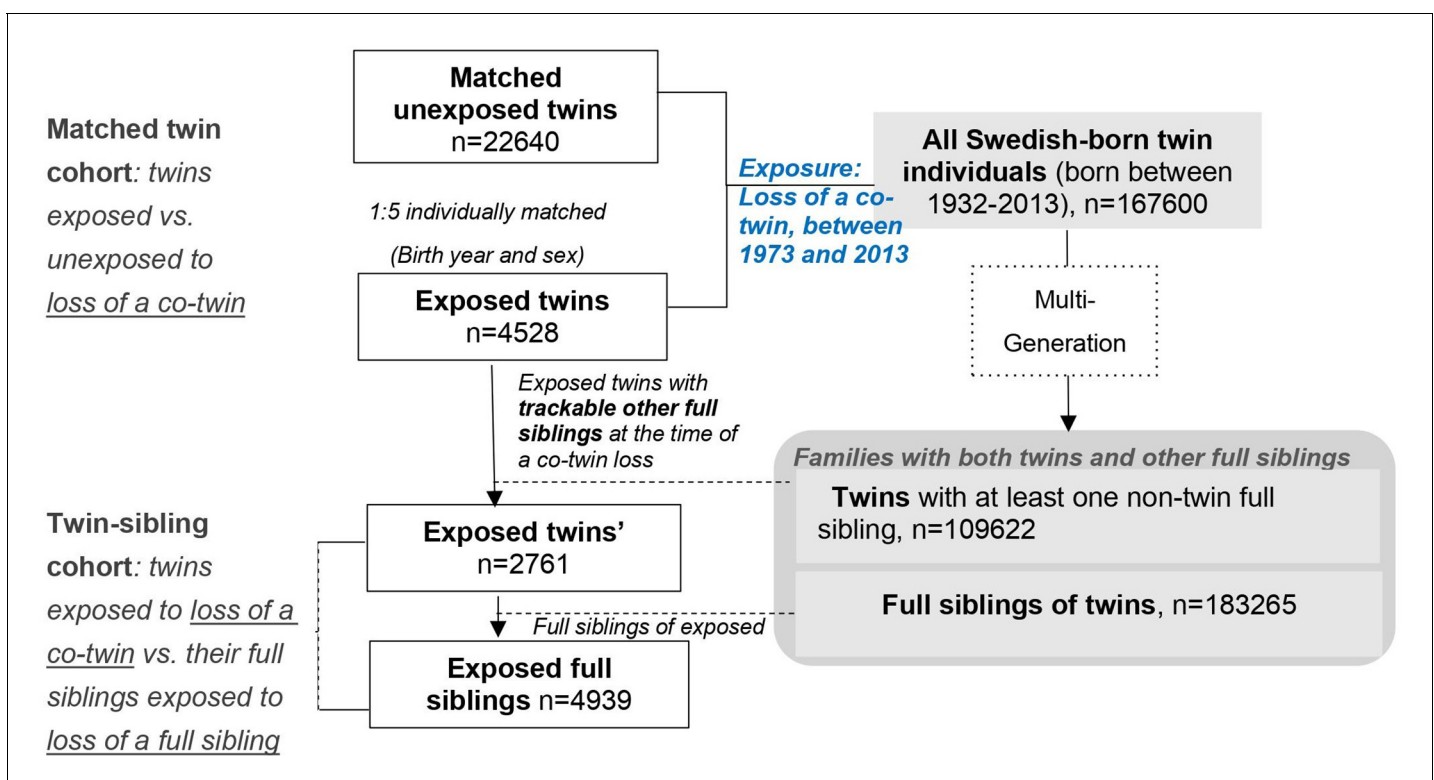

**Figure 1.** Study design.

controlling for familial factors (the twin-sibling family cohort). See details in 'Materials and methods', and *Figure 1* and *Supplementary file 1*-Table 1.

In total, the matched twin cohort (i.e., 4528 exposed twins and 22,640 matched unexposed twins) accrued 263,215 person-years, with an approximately 8 year average duration of follow-up. The median age at co-twin loss was 59 years and around 50% of the exposed twins were males (*Table 1*). While there were only small differences in family income, educational level, and marital status, exposed and unexposed twins differed somewhat with respect to history of severe somatic diseases and family history of psychiatric disorders (19.7% vs 11.8%, 49.9% vs 37.3%, respectively). As

**Table 1.** Characteristics of the study cohorts.

| | Matched twin cohort: twins exposed vs. unexposed to loss of a co-twin | | Twin-sibling cohort: twins exposed to loss of a co-twin vs. their full siblings | |
| --- | --- | --- | --- | --- |
| | Exposed twins | Matched unexposed twins | Exposed Twins' | Exposed full siblings |
| Number of individuals | 4528 | 22640 | 2761 | 4939 |
| Age at index date, median (IQR), year | 59(47-67) | 59(47-67) | 57(45-65) | 57(46-65) |
| Follow-up time, median (IQR), year | 7.4 (3.1–13.7) | 7.7 (3.2–14.2) | 8.0 (3.3–14.5) | 8.9 (3.8–15.4) |
| % of male | 50.8% | 50.8% | 51.7% | 49.7% |
| Education level, n (%) | | | | |
| <9 years | 1171 (25.9) | 5041 (22.3) | 700 (25.4) | 1187 (24.0) |
| 9–12 years | 2439 (53.9) | 12151 (53.7) | 1520 (55.1) | 2775 (56.2) |
| >12 years | 844 (18.6) | 5185 (22.9) | 505 (18.3) | 939 (19.0) |
| Unknown | 74 (1.63) | 263 (1.16) | 36 (1.30) | 38 (0.77) |
| Yearly family income level, n (%) | | | | |
| Lowest 20% | 504 (11.1) | 2275 (10.1) | 313 (11.3) | 561 (11.4) |
| Middle | 2487 (54.9) | 12092 (53.4) | 1499 (54.3) | 2653 (53.7) |
| Top 20% | 1050 (23.2) | 5850 (25.8) | 626 (22.7) | 1142 (23.1) |
| Unknown | 487 (10.8) | 2423 (10.7) | 323 (11.7) | 583 (11.8) |
| Marital status, n (%) | | | | |
| Single | 1270 (28.1) | 6073 (26.8) | 808 (29.3) | 1213 (24.6) |
| Married/cohabiting | 2624 (58.0) | 13739 (60.7) | 1564 (56.7) | 3056 (61.9) |
| Divorced/widowed | 634 (14.0) | 2828 (12.5) | 389 (14.1) | 670 (13.6) |
| History of severe somatic diseases[*], n (%) | | | | |
| Yes | 893 (19.7) | 2664 (11.8) | 514 (18.6) | 924 (18.7) |
| No | 3635 (80.3) | 19976 (88.2) | 2247 (81.4) | 4015 (81.3) |
| Family history of psychiatric disorders/suicide, n (%) | | | | |
| Yes | 2257 (49.9) | 8440 (37.3) | 1378 (49.9) | 2505 (50.7) |
| No | 2271 (50.2) | 14200 (62.7) | 1383 (50.1) | 2434 (49.3) |
| Cause of the co-twin's death, n (%) | | | | |
| *Unnatural death* | | | | |
| Yes | 1020 (22.5) | - | 666 (24.1) | 1213 (24.6) |
| No | 3508 (77.5) | - | 2095 (75.9) | 3726 (75.4) |
| Zygosity of twins, n (%) | | | | |
| Monozygotic twins | 746 (16.5) | 2367 (10.5) | 423 (15.4) | - |
| Dizygotic twins | 3016 (66.6) | 15722 (69.4) | 1851 (67.0) | - |
| Unknown | 766 (16.9) | 4551 (20.1) | 487 (17.6) | - |

* Involved somatic diseases included myocardial infarction, congestive heart failure, cerebrovascular disease, chronic pulmonary disease, connective tissue disease, diabetes, renal diseases, liver diseases, ulcer diseases, and HIV infection/AIDS.

expected, owing to the shared familial background, we observed more similarities between exposed twins and their full siblings on all studied characteristics (the 'twin-sibling cohort', *Table 1*).

During the follow-up, we identified 2047 twins with first-recorded psychiatric disorders in the matched twin cohort, including 526 among the exposed and 1521 among the unexposed twins, with a crude incidence rate (IR) of 12.29 and 6.90 per 1000 person-years, respectively (*Table 2*). The corresponding age- and sex-adjusted hazard ratio (HR) was 1.80 (95% confidence interval ⓘCI] 1.63–2.00), and decreased to 1.65 (95% CI 1.48–1.83) after further adjustment for socioeconomic factors, history of severe somatic disease, and family history of psychiatric disorders. Notably, the association was considerably stronger within the first month after a co-twin loss (HR = 7.16, 95% CI 3.07–16.70) than thereafter. The HR was 1.59 (95% CI 1.31–1.92) for the period 10 years and onwards. The twin-sibling cohort (i.e. exposed twins compared to their full siblings who also experienced a full sibling loss) yielded similar results, with a fully adjusted HR of 1.55 (95% CI 1.31–1.82) (*Table 2*).

Subgroup analyses, in both the matched twin and the twin-sibling cohorts, indicated that the association between loss of a co-twin and subsequent risk of psychiatric disorders did not differ by sex and family history of psychiatric disorders (*Supplementary file 1*-Table 2). Of note, although higher HRs were observed after loss of a co-twin due to unnatural death (e.g., death with any external cause), loss of a co-twin due to natural causes was also associated with a significantly increased risk of first-recorded psychiatric disorders among the surviving twins (e.g. HR = 1.49, 95% CI 1.32–1.69, based on analyses of matched twin cohort). In addition, an age-dependent risk pattern was found, suggesting that after age of 40, there was a decline in HR with increasing age at the index date (*Figure 2* and *Supplementary file 1*-Table 2).

To explore potential differences in relative risks by type of sibling loss, we did sub-analyses by zygosity (78% of included twins had such information). Compared to unexposed monozygotic twins, twins who lost a monozygotic co-twin were at a subsequently increased risk of psychiatric disorders (HR = 1.86, 95% CI 1.40–2.47), which was significantly higher than the risk elevation related to loss of a dizygotic co-twin among dizygotic twins (HR = 1.33, 95% CI 1.15–1.54, *P* for difference = 0.0395,

**Table 2.** Hazard ratios (HRs) with 95% confidence intervals (CIs) for any psychiatric disorder among the surviving twins after co-twin loss (overall and by follow-up time), compared to matched unexposed twins or their full siblings, derived from different Cox models.

| Model information | Matched twin cohort: twins exposed vs. unexposed to loss of a co-twin | | Twin-sibling cohort: twins exposed to loss of a co-twin vs. their full siblings | |
|---|---|---|---|---|
| | Number of cases (Crude incidence rate, per 1000 person years), exposed/unexposed twins | Hr (95% CI)* | Number of cases (Crude incidence rate, per 1000 person years), exposed twins/exposed siblings | Hr (95% CI)* |
| Model 1<br>Controlled/adjusted for sex, birth year | 526 (12.29)/1521 (6.90) | 1.80 (1.63–2.00) | 328 (11.92)/415 (7.90) | 1.54 (1.31–1.81) |
| Model 2<br>above + socioeconomic status (education level, marital status, family income) | | 1.79 (1.62–1.99) | | 1.55 (1.32–1.83) |
| Model 3<br>above + history of severe somatic disease | | 1.71 (1.55–1.90) | | 1.55 (1.31–1.82) |
| Model 4<br>above + family history of psychiatric disorder | | 1.65 (1.48–1.83) | | - |
| *By follow-up period*† | | | | |
| Within 1 month | 15 (43.47)/10 (5.79) | 7.16 (3.07–16.70) | 10 (47.55)/3 (7.96) | 7.21 (1.12–46.5) |
| 2–11 months | 45 (11.38)/116 (5.85) | 1.69 (1.18–2.42) | 29 (12.05)/30 (6.92) | 1.46 (0.83–2.57) |
| 2–9 years | 307 (12.24)/864 (6.75) | 1.61 (1.40–1.85) | 180 (11.39)/217 (7.30) | 1.58 (1.26–1.97) |
| 10 years and onwards | 159 (11.88)/531 (7.51) | 1.59 (1.31–1.92) | 109 (12.00)/165 (9.15) | 1.36 (1.03–1.81) |

\* Cox regression models were stratified by matching identifiers (birth year and sex, in matched twin cohort) or family identifiers (in twin-sibling cohort), and adjusted for covariates mentioned in the 'model information' column. Time since the index date was used as underlying time scale.
† HRs were derived from fully adjusted Cox regression models, that is model four for matched twin cohort and model three for twin-sibling cohort.

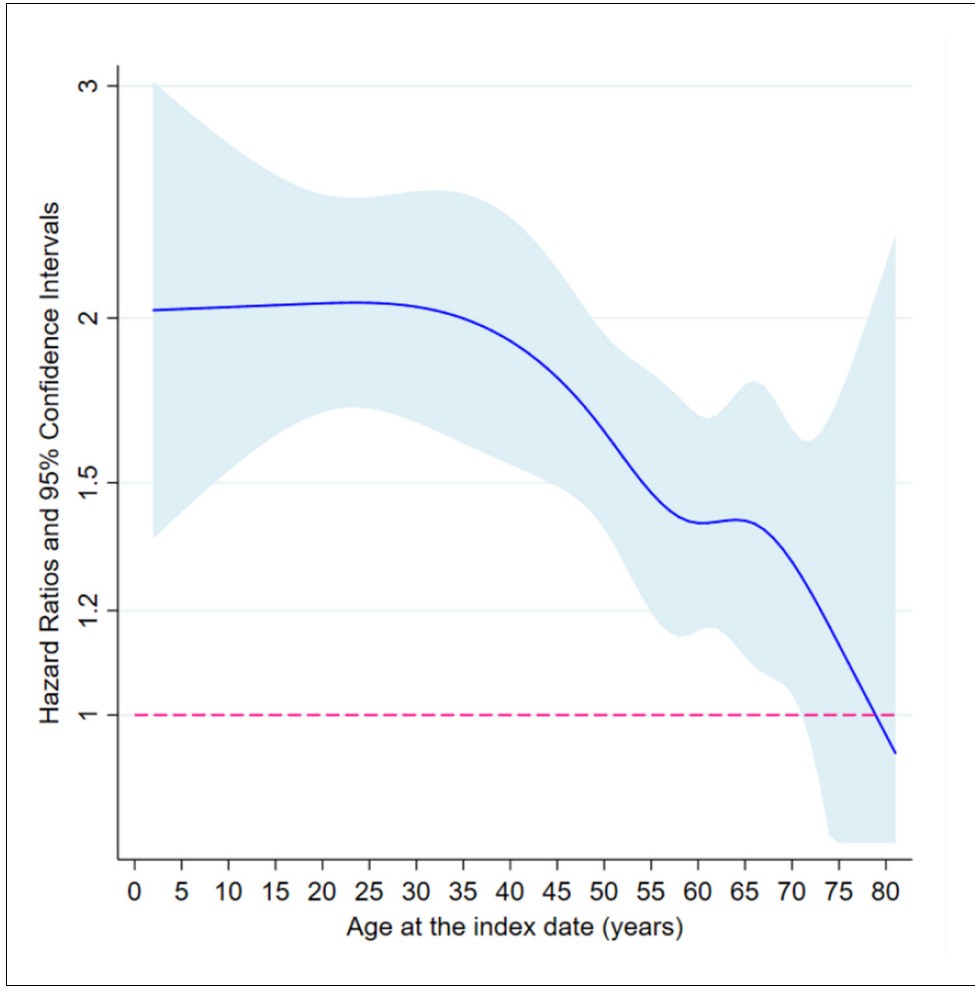

**Figure 2.** The association between loss of a co-twin and subsequent risk of psychiatric disorder by age at the index date, analyses of matched twin cohort. * Restricted cubic splines were applied on age at index date, with five knots placed at 5, 27.5, 50, 72.5, and 95 quantiles of the distribution of outcome events. Then, age-varying HRs were predicted based on fully adjusted Cox models where interaction terms between loss of a co-twin and splined age profiles were added. The cox models were stratified by matching identifiers, that is sex and birth year, and adjusted for education level, family income, marital status, history of severe somatic diseases, and family history of psychiatric disorders.

The online version of this article includes the following source data for figure 2:

**Source data 1.** Summary data for *Figure 2*.

*Table 3*). The analyses of twin-sibling cohort conveyed similar information, indicating a hierarchy of psychiatric disorder risk —the HR for 'loss of a monozygotic twin', 'loss of a dizygotic twin', and 'loss of a full sibling' was 2.45, 1.29, and 1.00, respectively (*Table 3*). In addition, we observed increasing HRs with larger age gaps between twins and non-twin full siblings held as reference (*P* for trend <0.001, *Supplementary file 1*-Table 2).

Based on the matched twin cohort, we separately estimated HRs for six subtypes of psychiatric disorders (*Figure 3*). The association between loss of a co-twin and risk of psychiatric disorders was noted for all subtypes of psychiatric disorders, with the highest HRs observed for psychotic disorders (HR 3.30, 95% CI 2.02–5.38) and stress-related disorders (HR 2.17, 95% CI 1.65–2.84) during the entire follow-up period, and for stress-related disorders (HR 5.36, 95% CI 2.31–12.40) and mood disorders (HR 2.63, 95% CI 1.32–5.24) during the first year after loss.

With slightly lower crude IRs, sensitivity analyses where the ascertainment of psychiatric disorders was merely based on the primary diagnoses in the National Patient Register (NPR) or the diagnoses from inpatient care in NPR revealed largely similar estimates (*Supplementary file 1*-Table 3).

**Table 3.** Hazard ratios (HRs) with 95% confidence intervals (CIs) for any psychiatric disorder among the surviving twins after co-twin loss, compared to matched unexposed twins (matched twin cohort) or their full siblings (twin-sibling cohort), by zygosity of the twin pairs

| | Twins who lost a monozygotic twin vs. matched unexposed monozygotic twins or their full siblings | | Twins who lost a dizygotic twin vs. matched unexposed dizygotic twins or their full siblings | |
| --- | --- | --- | --- | --- |
| | Number of cases (Crude incidence rate, per 1000 person years), exposed/unexposed | Hr (95% CI)* | Number of cases (Crude incidence rate, per 1000 person years), exposed/unexposed | Hr (95% CI)* |
| Matched twin cohort | 92 (14.81)/153 (7.26) | 1.86 (1.40–2.47) | 303 (10.89)/1024 (7.08) | 1.33 (1.15–1.54) |
| *By follow-up period* | | | | |
| Within 1 month | 6 (105.9)/2 (11.12) | 9.47 (1.88–47.8) | 8 (34.81)/5 (4.17) | 4.20 (1.15–15.3) |
| 2–11 months | 12 (18.7)/14 (6.84) | 2.54 (1.01–6.37) | 22 (8.37)/72 (5.23) | 1.23 (0.71–2.13) |
| 2–9 years | 46 (12.40)/93 (7.42) | 1.40 (0.95–2.07) | 190 (11.41)/603 (6.98) | 1.36 (1.13–1.63) |
| 10 years and onwards | 28 (15.56)/44 (6.99) | 2.50 (1.45–4.34) | 83 (10.00)/344 (7.97) | 1.22 (0.93–1.60) |
| *By cause of the co-twin's death* | | | | |
| Unnatural death | 28 (21.32)/26 (5.58) | 4.29 (2.18–8.45) | 81 (9.79)/254 (5.97) | 1.44 (1.09–1.91) |
| Natural death | 64 (13.07)/127 (7.74) | 1.46 (1.05–2.03) | 222 (11.36)/770 (7.54) | 1.30 (1.10–1.54) |
| *By gender difference* | | | | |
| Twins with same gender | 92 (14.81)/153 (7.26) | 1.86 (1.40–2.47) | 118 (10.51)/240 (6.93) | 1.43 (1.13–1.82) |
| Twins with different gender | - | - | 185 (11.15)/784 (7.13) | 1.33 (1.10–1.62) |
| Twin-sibling cohort | 53 (14.37)/57 (7.24) | 2.45 (1.56–3.85) | 193 (10.73)/279 (8.42) | 1.29 (1.05–1.59) |

\* Cox regression models were stratified by matching identifiers (birth year and sex, in matched twin cohort) or family identifiers (in twin-sibling cohort), and adjusted for education level, family income, marital status, history of severe somatic diseases, and family history of psychiatric disorder. Time since the index date was used as underlying time scale.

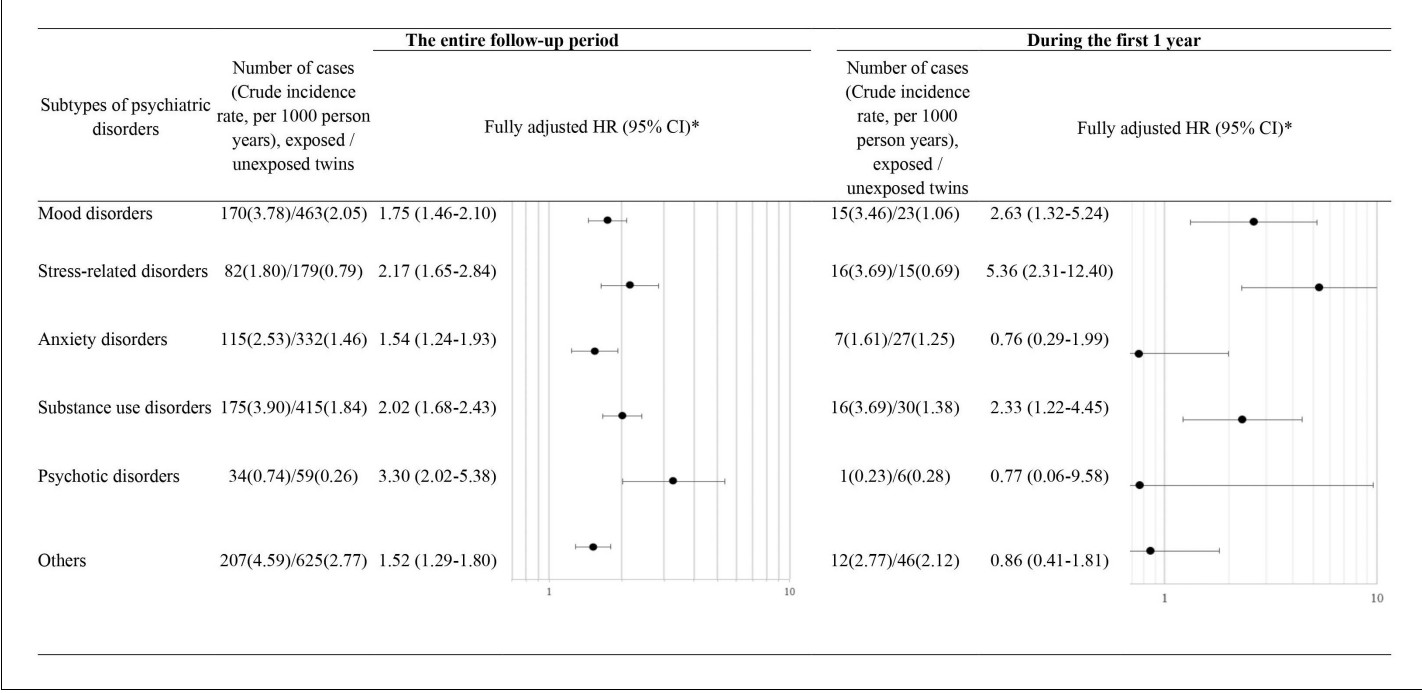

**Figure 3.** Hazard ratios (HRs) with 95% confidence intervals (CIs) for subtypes of psychiatric disorders among twins exposed to loss of a co-twin compared to matched unexposed twins (analyses of matched twin cohort), for the entire follow-up period and for the first year after the index date. * Cox regression models were stratified by matching identifiers (birth year and sex), and adjusted for education level, family income, marital status, history of severe somatic diseases, and family history of psychiatric disorder. Time since the index date was used as underlying time scale.

## Discussion

To our knowledge, this is the first nationwide population-based and sibling-matched cohort study testing the association between loss of a co-twin and the subsequent risk of psychiatric disorders. Compared to twins without such loss experience or full siblings of the bereaved twins (who also experienced a loss of a full sibling), twins exposed to a co-twin loss were at considerably elevated risk of psychiatric disorders, especially during the first month after loss and particularly when the loss occurred before the age of 40. The excess risk was most pronounced after loss of a monozygotic co-twin and was higher for loss of a dizygotic co-twin than loss of a full sibling. In addition, we observed evident risk elevation after co-twin loss through either natural or unnatural deaths.

The twin bond has been suggested to be the closest and most enduring human social relationship (*Segal et al., 1995*; *Segal, 2000*). Besides the genetic relatedness, the unique relationship between twins is characterized by distinct intimacy and ambiguous identity boundaries. The genetic relatedness, shared early life experiences, and attachment have been suggested to contribute to the development of the shared twin identity (*Schave, 1983*; *Macdonald, 2002*). Consequently, when exposed to a co-twin death, both the grief (i.e., emotional reactions) and confusion of identity may contribute to a profound and long-lasting vulnerability of surviving twins. Indeed, in line with previous literature describing intense grief of surviving twins after co-twin loss (*Segal and Ream, 1998*; *Rosendahl and Björklund, 2013*), in the present study, we showed that the subsequent risk of developing first-diagnosed psychiatric disorders increased by 55–65% in the twin population, after carefully controlling for familial factors. Of note, such risk elevation was most obvious during the first month (7-fold), but persistently existed more than 10 years after the loss. In addition to the aforementioned factors (i.e., grief and identity confusion) which might underlie both the short- and long-term rise in the risk of psychiatric disorders, the long-term mental health decline among the surviving twins could also be attributable to the impairment of beneficial social support due to the co-twin loss and possible life-style changes, such as substance abuse or sleep problems (*Sinha, 2001*).

Moreover, consistent with prior theories that twin attachment changes across developmental stages, we found particularly strong risk increases of first-recorded psychiatric disorders when twins were exposed to co-twin loss in childhood and early adulthood (*Neyer, 2002*; *Segal and Bouchard, 1993*). Indeed, young people seem more vulnerable to various health consequences of trauma. We have previously reported higher excess risk of life-threatening infections and cardiovascular disease among individuals with stress-related disorders after trauma at an earlier age (*Song et al., 2019*; *Song et al., 2018*). Long-lasting biological interruptions, for example promoted inflammatory reactions (*Danese et al., 2011*) and neuropsychological/cognitive impairment (*De Bellis and Zisk, 2014*; *Kennedy et al., 2018*), have been associated with childhood exposure to trauma.

Without any comparable data, our finding that individuals suffering the loss of a monozygotic co-twin were at higher risk of psychiatric disorders compared to individuals suffering loss of co-dizygotic twin gains support from previous reports indicating a greater grief intensity (*Segal and Bouchard, 1993*), together with a less grief reduction (*Segal and Ream, 1998*), after loss of a monozygotic co-twin relative to loss of a dizygotic co-twin. Interpretations for this phenomenon include evolutionary theories linking more intense grief after the bereavement to greater genetic relatedness to the deceased (*Neyer, 2002*; *Segal and Ream, 1998*; *Parkes, 1993*). Alternatively, monozygotic twins have been reported to be emotionally closer and more dependent on their co-twin than dizygotic twins (*Fortuna et al., 2010*; *Penninkilampi-Kerola et al., 2005*). Thus, it is plausible that losing a monozygotic co-twin can result in more severe psychiatric consequences, compared to losing a dizygotic co-twin.

The abovementioned explanations may also apply to our next finding indicating that the loss of a full sibling is associated with lower risk of psychiatric disorder compared to loss of a co-twin. Particularly, with respect to dizygotic twins, the twin pair shares equal genetic relatedness as a twin-sibling pair. Therefore, even with an extreme assumption that the increase in the risk of psychiatric disorders after loss of a monozygotic co-twin than loss of a dizygotic co-twin was solely due to genetic factors, the 29% increased rate of psychiatric disorders observed after loss of a dizygotic co-twin than loss of a full sibling should be largely attributable to bereavement. Furthermore, the stronger associations observed with larger age gaps between twins and non-twin siblings supports the notion that siblings closer in age who share more life experiences are more likely have a closer relationship

(*Conger and Little, 2010*) and hence might experience more difficult bereavement after a sibling loss. Similarly, other studies have also suggested a 15–20% excess risk of psychiatric disorders among individuals exposed to loss of a full sibling, as compared to the general population (*Guldin et al., 2017*; *Rostila et al., 2019*).

While we attempted to evaluate the influence of co-twin loss on the potential rise in psychiatric disorders, genetic confounding, namely the genetic susceptibility to psychiatric disorders and premature death shared within a twin pair, especially the monozygotic twins, is a concern in the present study. We made every effort to address this issue in the design of our study (e.g. excluding twins with past history of psychiatric disorders, controlling for family history of psychiatric disorders, and comparisons with full siblings) as well as in our analytic strategies (e.g. disentangling natural from unnatural deaths and performing a time-trend analysis). Although we cannot completely exclude the possibility of residual confounding, the results of the abovementioned analyses do argue against a pure explanation by genetic factors. Also, the time-trend analyses demonstrated – above and beyond possible genetic confounding - a distinct bereavement-response pattern, illustrated by the dramatic risk rise of first onset psychiatric disorders during the first months after the loss of a co-twin. Since we had limited information on prenatal/neonatal factors and behavior-related factors (e.g., smoking and alcohol consumption), residual confounding due to these factors remains another concern. Further research with detailed data on birth characteristics and lifestyle is warranted to clarify the role of such factors on the reported associations. Additionally, the absence of data from primary care as well as late inclusion of outpatient specialist care records in the NPR might have led to an underestimated number of individuals with milder forms of psychiatric disorders.

The major merit of our study is the use of population-based cohort design, including 4528 surviving twins with a complete follow-up of up to 40 years, and the between twin-full sibling comparison which enabled a direct comparison between two types of stressors, that is loss of a co-twin and loss of a full sibling, while controlling for potential familial confounders (e.g., genetic background was fully controlled for monozygotic twins, and partially controlled for dizygotic twins). As the largest cohort study on co-twin loss so far, we had sufficient statistical power to perform all planned subgroup analyses. Information bias was minimized since the registration and diagnosis of exposure and outcome was compiled prospectively and independently. Furthermore, the availability of rich information on sociodemographic and medical factors enabled considerations of a wide range of important confounding factors.

In conclusion, in the Swedish population, death of a co-twin was associated with a subsequently elevated rate of first-recorded psychiatric disorders among the surviving twins, particularly among monozygotic twins but also dizygotic twins as compared to their non-twin full siblings. Given that dizygotic twins share equal genetic relatedness to the twin-sibling pairs, this dose-response pattern suggests that both genetic relatedness and early life attachment may contribute to the subsequent risk of psychiatric disorders among surviving twins after co-twin loss.

## Materials and methods

From the Total Population Register, we identified all individuals born in Sweden between 1932 and 2013. We then linked these individuals to other registers in Sweden, including the Multi-Generation Register, the Causes of Death Register, the Swedish Twin Registry, and the National Patient Register (NPR), utilizing the personal identification numbers that are uniquely assigned to all Swedish residents.

From the Multi-Generation Register which includes almost complete familial link for all individuals born in Sweden since 1932, we identified all twin pairs as full siblings (i.e., with the same biological father and mother) that were born on the same day (+ / - 1 day), after excluding multiple births (more than two babies), and defined individuals with same biological father and mother but born on different dates as normal full siblings.

### Population-based matched twin cohort

As shown in *Figure 1*, among a total of the 167,600 identified twin individuals, 6732 experienced the loss of a co-twin due to death between 1973 and 2013, and thereby were included in the exposed cohort on the death date of their co-twin (i.e., the index date for exposed twins). To enable the establishment of attachment relationship (*Howe and Ross, 1990*), as well as to reduce the

possibility of a co-twin loss due to poor birth condition or congenital defects that shares within a twin pair, twins who lost a co-twin before age of 2 years were removed from the exposed cohort (n = 1,241). We further excluded twins who had a history of psychiatric disorders (n = 790) or emigrated (n = 173) before the index date, leaving 4528 eligible exposed twins for further analyses. Then, per each exposed twin, we randomly selected five unexposed twins (individually matched by sex and birth year, n = 22,640) who were free of psychiatric disorders and whose co-twins were alive on the index date for exposed twins (i.e., the index date for unexposed twin).

### Twin-sibling cohort

We further constructed a twin-sibling cohort where we compared rates of psychiatric disorders between twins exposed to co-twin loss and their full siblings, if any, who were also exposed to the same loss. Since both the surviving twin and other non-twin full siblings were affected by the same bereavement of a twin loss, this cohort enabled a direct comparison between two types of stressors, that is loss of a co-twin and loss of a full sibling, while controlling for familial factors. Through the Multi-Generation Register, we identified 4939 full siblings of 2761 twins exposed to a co-twin loss, who were alive and free of psychiatric disorders on the death date of the deceased twin (i.e., the index date for the exposed full siblings).

### Follow-up

Follow-up of all study participants started from the index date, until the occurrence of any or a specific type of psychiatric disorders, death, emigration, or the end of follow-up (December 31, 2013), whichever occurred first. The follow-up for unexposed twins in the twin cohort was additionally censored if a co-twin loss appeared after the index date.

### Loss of a co-twin due to death

We obtained information on death of a co-twin and its underlying causes from the Swedish Causes of Death Register, which is available electronically for register-based research since 1952. Unnatural death was defined as a death with any external cause (according to the 8-10th Swedish revisions of the International Classification of Diseases [ICD] codes, *Supplementary file 1*-Table 1) as one of the documented causes based on the Causes of Death Register.

### Psychiatric disorders

Any, first-ever, inpatient or outpatient hospital visit with a psychiatric disorder as one of the registered diagnoses during the follow-up was identified from the NPR (ICD eight and ICD nine codes: 290–315, ICD 10 codes: F00-F99). For sub-analyses on six different subtypes of psychiatric disorders (i.e., mood disorders, stress-related disorders, anxiety disorders, substance use disorders, psychotic disorders, and others), first-ever diagnosis of each specific disorder was also extracted from the NPR, according to corresponding ICD codes shown in *Supplementary file 1*-Table 1.

### Covariates

Information on zygosity of twins was mainly retrieved from the Swedish Twin Registry, and we additionally counted all twin pairs with sex discordance as dizygotic twins. We acquired data on education level, family income, and marital status at index date from the Longitudinal Integration Database for Health Insurance and Labor Market study. History of severe somatic diseases (including myocardial infarction, congestive heart failure, cerebrovascular disease, chronic pulmonary disease, connective tissue disease, diabetes, renal diseases, liver diseases, ulcer diseases, and HIV infection/ AIDS, with ICD codes listed in *Supplementary file 1*-Table 1) was also collected from the NPR. Family history of psychiatric disorders was defined as any diagnosis of or death due to psychiatric disorders, suicide, or suicide attempts among the first-degree relatives (i.e., biological parents and siblings) of the study participants, according to the NPR or Causes of Death Register. In all analyses, we used the most updated information before the index date for each covariate. The study was approved by the Regional Ethics Review Board in Stockholm, Sweden.

## Statistical analysis

We estimated the association between loss of a co-twin and risk of psychiatric disorders using hazard ratios (HRs) with 95% confidence intervals (CIs), derived from conditional Cox regression models. Time since the index date was applied as the underlying time scale.

In the matched twin cohort, twins exposed and unexposed to loss of a co-twin were compared. The models were stratified by matching identifiers (sex and birth year), and partially or fully adjusted for education level (<9 years, 9–12 years,>12 years, unknown), family income (top 20%, middle, low-est 20%), marital status (single, married or cohabiting, divorced or widow), history of severe somatic diseases (yes/no), and family history of psychiatric disorders (yes/no). In subgroup analyses, we calcu-lated the HRs by sex (male/female), age at index date (2–18 years, 19–52 years, 53–64 years,≥65 years), family history of psychiatric disorders (yes/no), the cause of the co-twin' s death (natural death and unnatural death), and time since index date (within 1 month, 2–11 months, 2–9 years,≥10 years). Throughout our analyses, we checked the proportional hazards assumption graphically and by Schoenfeld's partial residuals; and neither of them revealed any indication of violation of this assumption. We further examined the effect modification by age at index date on the interested association and visualized age-varying HRs using methods described previously (*Fe, 2001*). Addition-ally, we did separate analyses for loss of a monozygotic co-twin (among monozygotic twins) and loss of a dizygotic co-twin (among dizygotic twins). The statistical significance of the differences between sub-group HRs were assessed by introducing interaction terms to the Cox models or by Wald test.

In addition to considering all psychiatric disorder as one group, we did sub-analyses for six differ-ent types of psychiatric disorders. We further explored the types of psychiatric disorders most proxi-mal to or within one year after the bereavement.

Next, we repeated the main analyses in the twin-sibling cohort. Cox models were stratified by family identifiers, and adjusted for age at the index date and sex as well as all covariates used in the matched twin cohort. Given that the age difference between siblings may affect the degree of their shared experiences and their relationship, we further tested the impact of age difference on the risk of psychiatric disorder after a loss by calculating HRs by age gaps between twins and non-twin full siblings (0–2, 3–5, and >5 years intervals). To test the robustness of the observed association to the outcome definition, we re-ran the analyses by using either only the main diagnosis from the NPR or only the diagnosis from inpatient care of NPR for identifying psychiatric disorders. All analyses were conducted in SAS statistical software, version 9.4 (Cary, NC) and STATA 15 (StataCorp LP). SAS script used in the primary analyses are available (*Source code 1*).

## Acknowledgements

The Swedish Twin Registry is managed by Karolinska Institutet and receives funding through the Swedish Research Council.

## Additional information

### Funding

| Funder | Grant reference number | Author |
|---|---|---|
| Icelandic Centre for Research | 163362-051 | Unnur A Valdimarsdóttir |
| European Research Council | Consolidator Grant 726413 | Unnur A Valdimarsdóttir |
| Swedish Research Council | 340-2013-5867 | Catarina Almqvist |
| Swedish Research Council | 2017-00641 | Patrik KE Magnusson |
| Karolinska Institutet | Senior Researcher Award andStrategic Research Area in Epidemiology | Fang Fang |
| National Science Foundation of China | 81971262 | Huan Song |

The funders had no role in study design, data collection and interpretation, or the decision to submit the work for publication.

## Author contributions

Huan Song, Conceptualization, Data curation, Formal analysis, Validation, Investigation, Methodology, Writing - original draft, Writing - review and editing; Henrik Larsson, Resources, Data curation, Writing - review and editing; Fang Fang, Conceptualization, Supervision, Methodology, Writing - review and editing; Catarina Almqvist, Nancy L Pedersen, Supervision, Methodology, Writing - review and editing; Patrik KE Magnusson, Conceptualization, Resources, Data curation, Supervision, Writing - review and editing; Unnur A Valdimarsdóttir, Conceptualization, Resources, Data curation, Formal analysis, Funding acquisition, Methodology, Writing - review and editing

## Author ORCIDs

Huan Song (iD) https://orcid.org/0000-0003-3845-8079
Unnur A Valdimarsdóttir (iD) https://orcid.org/0000-0001-5382-946X

## Ethics

Human subjects: The study was approved by the Regional Ethics Review Board in Stockholm, Sweden (Dnr 2013/862-31/5); and the requirement of informed consent was waived for register-based studies in Sweden.

## Decision letter and Author response

Decision letter https://doi.org/10.7554/eLife.56860.sa1
Author response https://doi.org/10.7554/eLife.56860.sa2

# Additional files

## Supplementary files

• Source code 1. SAS script for the primary analyses.

• Supplementary file 1. Supplementary Tables. Supplementary Table 1. International Classification of Diseases (ICD) codes, eighth (ICD-8; 1969–1986), ninth (ICD-9; 1987–1996), and tenth (ICD-10; 1997–2013) Swedish revisions for diagnoses used in this study Supplementary Table 2. Hazard ratios (HRs) with 95% confidence intervals (CIs) for any psychiatric disorder among the surviving twins after co-twin loss, compared to matched unexposed twins or their full siblings, by different characteristics Supplementary Table 3. Hazard ratios (HRs) with 95% confidence intervals (CIs) for any psychiatric disorder, identified by only primary diagnosis in the National Patient Register or by only diagnosis from National Inpatient Register, among the surviving twins after co-twin loss, compared to matched unexposed twins (matched twin cohort) or their full siblings (twin-sibling cohort).

• Transparent reporting form

## Data availability

Two source data files have been provided. Original data is held by Swedish National Board of Health and Welfare, Statistics Sweden and the Swedish Twin Registry. Due to Swedish law on data protection and the ethical approval of the current study, we cannot make the data publicly available. However, any researcher can access the data by obtaining an ethical approval from a regional ethical review board and thereafter request the original data from the Swedish National Board of Health and Welfare, Statistics Sweden, and the Swedish Twin Register. Detailed information on data application can be found at https://www.registerforskning.se/en/ and https://ki.se/en/research/the-swedish-twin-registry.

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
