## [Decision Letter]

**Acceptance summary:**

Using the excellent Swedish registry system, this manuscript shows that being bereaved of a co-twin puts you at an increased risk of being diagnosed with a psychiatric disorder later in life. This is an interesting and novel finding that contributes to the literature on health consequences of sibling loss.

**Decision letter after peer review:**

Thank you for submitting your article "Risk of psychiatric disorders among the surviving twins after a co-twin loss" for consideration by *eLife*. Your article has been reviewed by three peer reviewers, and the evaluation has been overseen by a Reviewing Editor and Eduardo Franco as the Senior Editor. The following individual involved in review of your submission has agreed to reveal their identity: Maria H. Chahrour (Reviewer #3).

The reviewers have discussed the reviews with one another and the Reviewing Editor has drafted this decision to help you prepare a revised submission.

Summary:

This manuscript aimed to estimate the association between loss of a co-twin and the risk psychiatric disorder using a nationwide population-and sibling-matched cohort study. Leveraging the excellent Swedish registry system, they analyzed data from a cohort of ~4,500 twins who had lost a co-twin, their ~4,900 sibs, and ~23,000 non-bereaved twins. The authors reported that being bereaved of a co-twin puts you at an increased risk of being diagnosed with a psychiatric disorder later in life. The relative risk of the bereaved twins was increased compared to their siblings, after loss of a monozygotic (the highest risk) or a dizygotic co-twin. This well-written manuscript, which used a high quality and unique twin data, is a nice addition to the literature on mental health consequences by sibling loss. However, there are a few concerns.

Essential revisions:

• While studying monozygotic twins and comparing them to dizygotic ones may give indications about the significance of relationship for health consequences by bereavement, it also opens up the possibility that much of the association is due to genetic confounding and a genetic predisposition for psychiatric health problems especially because the strongest association is found in monozygotic twins. The authors should elaborate more on this possibility and also be critical about whether associations reflect a bereavement effect or confounding/underlying risk factors (residual confounding). The possibility of confounding should be highlighted in the Introduction of the paper. In addition, the authors should discuss the analytical strategies used to reduce and estimate the risk of confounding in the Discussion section. For example, to what extent does the authors believe that their results indicate a "bereavement effect"?

• The authors could be clearer about the expected mental health consequences by twin loss compared to non-twin sibling loss by comparing these two events. Why is twin loss more detrimental? A possibility is to empirically compare the two types of sibling losses. Is it possible to study how twin loss influence bereaved twin/non-twin siblings within the same family? Then, it would be necessary to study families with twins where there is another non-twin sibling (or several non-twins). Twin siblings are also the exact same age and the association could therefore reflect that the siblings have a closer relationship because they naturally have shared their lives with one another. Is the loss of non-twin siblings with a small age differences more detrimental?

• Previous research has suggested closer relationships between female siblings and that women have higher quality relationships with family members across the life course. Why did the authors decide not to focus on the age at loss? or gender differences? How about having additional siblings?

• A paper studying the risk of mental health problems following sibling loss is missing. Rostila et al., 2019. It could be interesting to compare the results from the two studies.

• The authors study short term and long term effects by twin loss. However, what could explain these short or long term consequences on psychiatric disorder? Complicated grief and post-traumatic stress could be, for instance, underlie short-term effects.

• How can the stronger association by loss of monozygotic compared to loss of dizygotic twin be explained? They should have an equally close relationship given that they follow each other across life from birth to death. This should be discussed much more thoroughly in the paper and also the comparison to non-twin siblings.

• The different follow-up periods and the rationale behind these are not mentioned in the Introduction or the Results but are discussed in the Discussion section. The Discussion section also includes other results that are not previously presented in the paper. Please check so that new findings are discussed.

• Twins who lost a co-twin before the age of 2 were removed in order to establish an attachment relationship. Why did the authors use this cut off?

• The authors used the Cox proportional hazard modelling; however, they did not mention if the proportional hazard assumption was met for any of the models. While this does not invalidate their findings, it should be described in the Materials and methods and reported in the Results. The proportional hazard assumption checking is probably implied with the age-time interaction spline modelling (see Hess, https://doi.org/10.1002/sim.4780131007) that was used in the Materials and methods and shown in Figure 2 (with the HR dropping off from age 40). However, the authors need to explicitly connect this to proportional hazard. Even better, they could formally check these assumptions using K-M curves or the Schonfield residuals.

• The paper would benefit from a comprehensive discussion of the implications of their finding; the hazard ratios drops off precipitously from when the surviving twin hits the age of 40. This was given scant to no attention in the Discussion section.

---

## [Author Response]

Essential revisions:• While studying monozygotic twins and comparing them to dizygotic ones may give indications about the significance of relationship for health consequences by bereavement, it also opens up the possibility that much of the association is due to genetic confounding and a genetic predisposition for psychiatric health problems especially because the strongest association is found in monozygotic twins. The authors should elaborate more on this possibility and also be critical about whether associations reflect a bereavement effect or confounding/underlying risk factors (residual confounding). The possibility of confounding should be highlighted in the Introduction of the paper. In addition, the authors should discuss the analytical strategies used to reduce and estimate the risk of confounding in the Discussion section. For example, to what extent does the authors believe that their results indicate a "bereavement effect"?

We certainly agree that possibility of genetic confounding needs to be adequately addressed in our study. In our original manuscript, several approaches have been used for controlling for familial factors:

1) When designing the study, we added a twin-sibling cohort where we compared rates of psychiatric disorders between twins exposed to co-twin loss and their full siblings (i.e., a comparison within a family). Since both the surviving twin and their non-twin full siblings were affected by the same bereavement of a twin loss, this cohort enabled a direct comparison between two types of stressors, i.e., loss of a co-twin and loss of a full sibling, while controlling for shared familial factors between siblings. Particularly, with regard to dizygotic twins, the twin pair shares equal genetic relatedness as the twin-sibling pair. Therefore, even with an extreme assumption that the increase in the risk of psychiatric disorders after loss of a monozygotic co-twin than loss of a dizygotic co-twin was solely due to a greater genetic relatedness to the deceased, the 29% increased rate of psychiatric disorders observed after loss of a dizygotic co-twin than loss of a full sibling, as shown in Table 3, should be largely attributable to bereavement. Importantly, given the reference group here is also a cluster of bereaved individuals (due to loss of a full sibling), who had also an 15-20% increased risk of developing psychiatric disorder after a loss of sibling(according to literature) , the ‘29% increase’ is reasonably a conservative estimate.

2) The sub-analyses focusing on the cause of the co-twin’s death demonstrated a clearly elevated rate of psychiatric disorders among twins losing a co-twin due to natural death (increase by 49% compared to matched unexposed twins, by 62% compared to their full siblings). Given that a natural death is less likely affected by a liability to psychiatric disorders shared by the twin pair, this result provides some relief to the concern that the observed association is mainly driven by the shared genetic liability.

3) Importantly, all analyses were limited to twins or sibling pairs without previous history of psychiatric disorders. Additional adjustment for family history of psychiatric disorders (in the twin cohort) attenuated the point estimates somewhat but marginally – see Table 2, the adjusted HR decreased from 1.71 to 1.65 after additionally adjusting for family history of psychiatric disorder).

4) The results from our time-trend analysis indicate considerable “emotional shock” effect as observed by the considerable elevated HRs during the first month after loss of a co-twin (HR=7.16, see Table 2). This phenomenon held true both after loss of monozygotic twins (HR=9.47, see Table 3) and dizygotic twins (HR=4.20), also in comparison with their full bereaved sibling (HR=7.21, see Table 2). If only genetic factors contributed to the increased risks of psychiatric disorders after co-twin loss, we would have expected a monotone rise across time after the co-twin loss.

The analytic strategies we used were stated in our Discussion section. We have now highlighted the importance of controlling for genetic confounding in the Introduction part, and thoroughly discussed this concern in the Discussion of the revised manuscript.

In the revised manuscript, we added:

‘Introduction section: “Moreover, given the shared genetic background of twins, the psychiatric consequences after a co-twin loss need to be estimated with caution, ideally using family comparison with additional vigorous control of previous psychiatric morbidities and the causes of the co-twin’s death.”

Discussion section: “Particularly, with respect to dizygotic twins, the twin pair shares equal genetic relatedness as a twin-sibling pair. Therefore, even with an extreme assumption that the increase in the risk of psychiatric disorders after loss of a monozygotic co-twin than loss of a dizygotic co-twin was solely due to genetic factors, the 29% increased rate of psychiatric disorders observed after loss of a dizygotic co-twin than loss of a full sibling should be largely attributable to bereavement.”

Discussion section: “While we attempted to evaluate the influence of co-twin loss on the potential rise in psychiatric disorders, genetic confounding, namely the genetic susceptibility to psychiatric disorders and premature death shared within a twin pair, especially the monozygotic twins, is a concern in the present study. […] Also, the time-trend analyses demonstrated – above and beyond possible genetic confounding – a distinct bereavement-response pattern illustrated by the dramatic risk rise of first-onset psychiatric disorders during the first months after the loss of a co-twin.”

Discussion section: “’Since we had limited information on prenatal/neonatal factors and behavior-related factors (e.g., smoking and alcohol consumption), residual confounding due to these factors remains another concern.”

• The authors could be clearer about the expected mental health consequences by twin loss compared to non-twin sibling loss by comparing these two events. Why is twin loss more detrimental? A possibility is to empirically compare the two types of sibling losses. Is it possible to study how twin loss influence bereaved twin/non-twin siblings within the same family? Then, it would be necessary to study families with twins where there is another non-twin sibling (or several non-twins).

Thank you for this important comment. We fully agree that studying how twin loss influence bereaved twin vs. non-twin siblings within the same family is very informative on our hypothesis. This is exactly why we performed the twin-sibling cohort where bereaved twins are compared with bereaved non-twin full siblings within the same family.

The finding that loss of a co-twin led to a higher risk of psychiatric disorders compared to loss of a full sibling (HR=1.55, 95% CI 1.31-1.82), can both be explained by greater genetic relatedness, in case of monozygotic twins, and a closer relationship between a twin pair than a pair of full sibling.

In our opinion, probably one of the strongest indications for a “the twin-related bereavement effect” in our data is that bereaved dizygotic twins have 29% increased risk of first-onset psychiatric disorders compared to their bereaved full siblings. As dizygotic twins share equal genetic relatedness to the deceased twin as their full siblings, this finding indicates that the co-twin bereavement constitutes an additional burden to full-sibling loss.

Please refer to the statement in the manuscript:

Materials and methods section: “We further constructed a twin-sibling cohort where we compared rates of psychiatric disorders between twins exposed to co-twin loss and their full siblings, if any, who were also exposed to the same loss. Since both the surviving twin and other non-twin full siblings were affected by the same bereavement of a twin loss, this cohort enabled a direct comparison between two types of stressors, i.e., loss of a co-twin and loss of a full sibling, while controlling for familial factors.”

We discuss this issue in more detail when exploring potential mechanisms for the observed hierarchy in the rate elevation of psychiatric disorders following ‘loss of a monozygotic twin’, ‘loss of a dizygotic twin’, and ‘loss of a non-twin full sibling’ in the revised manuscript:

Discussion section: “Without any comparable data, our finding that individuals suffering the loss of a monozygotic co-twin were at higher risk of psychiatric disorders compared to individuals suffering loss of co-dizygotic twin gains support from previous reports indicating a greater grief intensity (Segal and Bouchard, 1993), together with a less grief reduction (Segal and Roam, 1998), after loss of a monozygotic co-twin relative to loss of a dizygotic co-twin. […] Thus, it is plausible that losing a monozygotic co-twin can result in more severe psychiatric consequences, compared to losing a dizygotic co-twin.”

Discussion section: “The abovementioned explanations may also apply to our next finding indicating that the loss of a full sibling is associated with lower risk of psychiatric disorder compared to loss of a co-twin. […] Furthermore, the stronger associations observed with larger age gaps between twins and non-twin siblings supports the notion that siblings closer in age who share more life experiences are more likely have a closer relationship (Conger and Little, 2010) and hence might experience more difficult bereavement after a sibling loss.”

Twin siblings are also the exact same age and the association could therefore reflect that the siblings have a closer relationship because they naturally have shared their lives with one another. Is the loss of non-twin siblings with a small age differences more detrimental?

This is an excellent point. We have now performed subgroup analyses by age difference (i.e., birth year difference between twins and bereaved non-twin full siblings) in the twin-sibling cohort. Please find the results in Supplementary file 1—supplementary table 2. The results indicate that the difference in rates of first-onset psychiatric disorders between bereaved twins and non-twin siblings becomes greater with increasing age differences. Namely, the bereaved non-twin full siblings who were at a similar age of the deceased twin suffered more similar risk of psychiatric disorder as the surviving twin after loss. This finding supports the notion, as suggested by the reviewer, that the excess risk in psychiatric disorders among surviving twins is partly due to the proximity in age (indicating more shared experiences and, perhaps, closeness of relationship).

We have added this analysis and the corresponding result to the revised manuscript.

Abstract: “Similarly, compared to non-twin full siblings, the adjusted relative risks were significantly increased after loss of monozygotic co-twin (2.45-fold), and loss of a dizygotic co-twin (1.29-fold), with higher HR observed with greater age gaps between twins and non-twin siblings.”

Materials and methods section: “Given that the age difference between siblings may affect the degree of their shared experiences and their relationship, we further tested the impact of age difference on the risk of psychiatric disorder after a loss by calculating HRs by age gaps between twins and non-twin full siblings (0-2, 3-5, and >5 years intervals).”

Results section: “In addition, we observed increasing HRs with larger age gaps between twins and non-twin full siblings held as reference (P for trend <0.001, in Supplementary file 1—supplementary table 2).”

Discussion section: “Furthermore, the stronger associations observed with larger age gaps between twins and non-twin siblings supports the notion that siblings closer in age who share more life experiences are more likely have a closer relationship (Conger and Little, 2010) and hence might experience more difficult bereavement after a sibling loss.”

• Previous research has suggested closer relationships between female siblings and that women have higher quality relationships with family members across the life course. Why did the authors decide not to focus on the age at loss? or gender differences? How about having additional siblings?

Thank you for this comment. We did analyses by age and gender, shown in Supplementary file 1—supplementary table 2 (also quoted in Author response table 1). Briefly, we didn’t see a difference in HR by gender, but there was a clear trend of increasing HRs with lower age at loss.

Regarding the gender difference of dizygotic twin pairs, we did the analyses and showed the results in Table 3 in the manuscript. The HR was 1.43 (95% CI 1.13-1.82) and 1.33 (95% CI 1.10-1.62) after loss of a same-gender dizygotic twin and an opposite-gender dizygotic twin, respectively, implying no strong evidence for effect modification by gender.

According to the reviewer’s comments, we additionally performed sub-analyses for exposed twins with and without additional full sibling(s) at the time of loss and found similar estimates. Also, please note that exposed twins with addition full siblings were exactly the ones involved in the twin-sibling cohort. The crude incidence rates of psychiatric disorders among all exposed twins (12.29 per 1000 person-year) and exposed twins with additional full siblings (11.92 per 1000 person-year) have been shown in Table 2 in the original manuscript. We therefore didn’t add these results to the revised manuscript. But we are certainly willing to reconsider it on the editor/reviewer’s request.

Author response table 1. Hazard ratios (HRs) with 95% confidence intervals (CIs) for any psychiatric disorder among the surviving twins after co-twin loss, compared to matched unexposed twins or their full siblings, by different characteristics

^*^ Cox regression models were stratified by matching identifiers (birth year and sex, in matched twin cohort) or family identifiers (in twin-sibling cohort), and adjusted for covariates mentioned in the ‘model information’ column. Time since the index date was used as underlying time scale.

• A paper studying the risk of mental health problems following sibling loss is missing. Rostila et al., 2019. It could be interesting to compare the results from the two studies.

Thank you for your comments. We have added a sentence, discussing the results of the mentioned paper.

In the revised manuscript:

‘Discussion section: “Similarly, other studies have also suggested a 15-20% excess risk of psychiatric disorders among individuals exposed to loss of a full sibling, as compared to the general population (Guldin et al., 2017; Rostila et al., 2019).”

• The authors study short term and long term effects by twin loss. However, what could explain these short or long term consequences on psychiatric disorder? Complicated grief and post-traumatic stress could be, for instance, underlie short-term effects.

As the reviewer suggested, we have now discussed in more details the underlying possible mechanisms for short-term and long-term effects of co-twin loss on the risk of psychiatric disorders.

In the revised manuscript:

Discussion section: “In addition to the aforementioned factors (i.e., grief and identity confusion) which might underlie both the short- and long-term rise in the risk of psychiatric disorders, the long-term mental health decline among the surviving twins could also be attributable to the impairment of beneficial social support due to the co-twin loss and possible life-style changes, such as substance abuse or sleep problems (Sinha, 2001).”

• How can the stronger association by loss of monozygotic compared to loss of dizygotic twin be explained? They should have an equally close relationship given that they follow each other across life from birth to death. This should be discussed much more thoroughly in the paper and also the comparison to non-twin siblings.

Thank you for this comment. We have now added a new paragraph in the revised manuscript, discussing the possible explanations for a stronger association observed after loss of a monozygotic twin than after loss of a dizygotic twin.

Discussion section: “Without any comparable data, our finding that individuals suffering the loss of a monozygotic co-twin were at higher risk of psychiatric disorders compared to individuals suffering loss of co-dizygotic twin gains support from previous reports indicating a greater grief intensity (Segal and Bouchard, 1993), together with a less grief reduction (Segal and Roam 1998), after loss of a monozygotic co-twin relative to loss of a dizygotic co-twin. […] Thus, it is plausible that losing a monozygotic co-twin can result in more severe psychiatric consequences, compared to losing a dizygotic co-twin.”

• The different follow-up periods and the rationale behind these are not mentioned in the Introduction or the Results but are discussed in the Discussion section. The Discussion section also includes other results that are not previously presented in the paper. Please check so that new findings are discussed.

Thank you. We have closely checked and ensured all estimates presented in the Discussion section have been properly presented in the Results section.

• Twins who lost a co-twin before the age of 2 were removed in order to establish an attachment relationship. Why did the authors use this cut off?

Thank you for your question. Our reasons of using 2 years as a cut-off point include:

a) Although the attachment bond between children and their parents has been reported to be formed as early as 8-12 months, the social interaction relevant to sibship-relationship formation depends on the activation of exploratory system, which may start around the age of 2-3. (https://hal-univ-tlse2.archives-ouvertes.fr/hal-01498767/document)

b) Given that a very early death of a twin baby (e.g., infant mortality) may be due to difficult birth or congenital defects, which can be also highly related to the physical condition of the surviving twin (including perhaps also psychiatric disorders), we chose to exclude loss of co-twin at very early age (less than 2 years) as exposure in the present study.

In our study population, among a co-twin loss within the first two years of life, more than 70% of the losses occurred within 60 days after twin birth. We found 787 co-twin deaths within 60 days, 332 between 61 days and 1 year, and 122 between 1 and 2 years after twin birth. Therefore, restricting to ‘1 year old’ age at loss gains little additional data power.

In the revised manuscript, we added:

Materials and methods section: “To enable the establishment of attachment relationship (Dufoyer, 1980) , as well as to reduce the possibility of a co-twin loss due to poor birth condition or congenital defects that shares within a twin pair, twins who lost a co-twin before age of 2 years were removed from the exposed cohort (n=1,241).”

• The authors used the Cox proportional hazard modelling; however, they did not mention if the proportional hazard assumption was met for any of the models. While this does not invalidate their findings, it should be described in the Materials and methods and reported in the Results. The proportional hazard assumption checking is probably implied with the age-time interaction spline modelling (see Hess, https://doi.org/10.1002/sim.4780131007) that was used in the Materials and methods and shown in Figure 2 (with the HR dropping off from age 40). However, the authors need to explicitly connect this to proportional hazard. Even better, they could formally check these assumptions using K-M curves or the Schonfield residuals.

Thank you for pointing out this important point. During the analyses, we checked the proportional hazards assumption graphically and by Schoenfeld’s partial residuals and found no evidence of assumption violation. Please note that since the exposed and unexposed twins were individually matched by age at the index date (i.e., therefore, age at loss was stratified in the COX model as a matching identifier), the change of HRs with age at the index does not imply an violence of proportional hazard assumption.

In the revised manuscript, we added:

Materials and methods section: “Throughout our analyses, we checked the proportional hazards assumption graphically and by Schoenfeld’s partial residuals; and neither of them revealed any indication of violation of this assumption”.

• The paper would benefit from a comprehensive discussion of the implications of their finding; the hazard ratios drops off precipitously from when the surviving twin hits the age of 40. This was given scant to no attention in the Discussion section.

Thank you for your comment. In the revised manuscript, we discussed the more pronounced association observed among individuals that experienced a co-twin loss at an earlier age and have now added this to the Discussion.

Discussion section: “Moreover, consistent with prior theories that twin attachment changes across developmental stages, we found particularly strong risk increases of first-recorded psychiatric disorders when twins were exposed to co-twin loss in childhood and early adulthood (Neyer, 2002; Segal and Bouchard, 1993). […] Long-lasting biological interruptions, e.g., promoted inflammatory reactions (24) and neuropsychological/cognitive impairment (25, 26), have been associated with childhood exposure to trauma.”